# An attribute-enhanced relationship-aware neighborhood matching model with dual attention

**Junlin Gu[1], Weiwei Liu[1], Xiong Yang** [2]*

**1** Department of Computer, Jiangsu Vocational College of Electronics and Information, Huai'an, China,
**2** Department of Computer Engineering, Fuzhou University Zhicheng College, Fuzhou, China

* 02116828@fdzcxy.edu.cn

**Data availability statement:** The data underlying the results presented in the study are available from https://www.nbsdc.cn/general/dataLinks/16666.11.nbsdc.6MnfkGWL.

## Abstract

The entity alignment task aims to match semantically corresponding entities in different knowledge graphs, which is important for knowledge fusion. Traditional graph-based methods often lose information due to insufficient use of attributes and imperfect relationship modeling, which makes it difficult to capture the deep semantic relationship between entities fully. To improve the effect of entity alignment, we propose a new model named ARNM-DAE2A, which strengthens the information aggregation capability of GCN by introducing a dual-attention mechanism to ensure a more balanced and comprehensive structural representation. The model contains the entity structure embedding module, the attribute structure embedding module, the joint alignment module and the relationship-aware neighborhood matching module. The entity structure embedding module optimizes the structure learning capability of GCN by introducing the pairwise attention mechanism. The attribute structural embedding module utilizes GCN to acquire entity attribute information. The joint alignment module weights and fuses the relationship structure information and attribute information as a comprehensive representation of entities. The relationship-aware neighborhood matching module then corrects the noise in the GCN aggregated information by comparing the neighborhood relationships of entity pairs. Experiments conducted on DBP15K and SRPRS datasets illustrate that the proposed ARNM-DAE2A outperforms baselines.

## 1 Introduction

Knowledge graph (KG) has shown great potential and value as a powerful form of knowledge representation and organization in various fields [1]. A KG is a semi-structured data model that graphically represents relationships among entities, including individuals, locations, occurrences, and various other elements [2]. It not only offers rich semantic information but also aids machines in comprehending and reasoning about the connections between entities. Despite presenting a wealth of semantic information, real-world hosts diverse KGs sourced from varying data sources, domains, and are described using different ontologies [3].

**Funding:** JG is funded by the Jiangsu Provincial Department of Industry and Information Technology Key Technology Innovation Project Guidance Plan under Grant 141-62-65, the Digital Public Service Platform Project of Jiangsu Provincial Department of Science and Technology under Grant 93208000931, and the Industry-university-research Project of Jiangsu Provincial Department of Science and Technology under Grant 20221343. XY is funded by Fujian Provincial Financial Research General Funding Project under Grant 2023CZ50. All funders play an important role in the study design, data collection and analysis, decision to publish, and preparation of the manuscript.

**Competing interests:** The authors have declared that no competing interests exist.

This heterogeneity poses substantial challenges for knowledge sharing and cross-graph applications.

Entity alignment refers to the process of discovering and matching mutually corresponding entities across different KGs or data sources [1,3]. Specifically, when there exist heterogeneous data describing similar concepts or entities in different graphs, the goal of entity alignment is to establish semantic correlations between these entities, enabling effective integration of these heterogeneous data on a semantic level. This implies that through entity alignment, entities with similar semantic meanings described in different data sources can be identified and mapped to consistent identifiers or representations [4]. Such mappings help eliminate semantic gaps between heterogeneous data, enhancing the quality and accuracy of data integration [5]. Through entity alignment, we can conduct knowledge queries, reasoning, and analysis in a cross-graph environment, maintaining semantic consistency while integrating heterogeneous data. This forms the basis for cross-domain, cross-data-source knowledge discovery, fostering deeper data collaboration and sharing. Entity alignment is not only a part of graph research but also a crucial tool driving cross-disciplinary knowledge fusion and application, laying the foundation for comprehensive, integrated intelligent information retrieval, and application [6].

In the context of entity alignment, the utilization of attribute information has become an indispensable aspect to capture semantic relationships and similarities between entities more accurately [7]. Entity attributes often encompass rich semantic features that can help differentiate entities to a certain extent. However, despite the wealth of semantic features in attribute information, existing entity alignment methods often overlook the correlations among attributes when utilizing them, leading to information loss and inaccuracy. Inherent connections and dependencies often exist among entity attributes, and these associations might be neglected in traditional approaches. Consequently, how to fully exploit attribute information and effectively capture the correlations between attributes has become a pressing challenge [8,9].

At the same time, to better model the relationships be-tween entities in entity alignment tasks, attention mechanisms have been incorporated into numerous entity alignment models. Traditional attention mechanisms often consider one-way information propagation and disregard the symmetry between entities, potentially resulting in incomplete information transfer, especially in entity pairs with symmetrical relationships. To comprehensively capture entity relationships, the dual attention mechanism has garnered significant attention in recent years [10,11]. The dual attention allows models to aggregate information from the perspectives of both entities simultaneously, aiding in a better understanding of their mutual influence. Through dual attention, the model can consider relationships between two entities simultaneously and map this relationship into a shared attention representation, thereby accurately capturing semantic connections between entities. However, the application of dual attention in the field of entity alignment remains relatively limited. While this mechanism has achieved some success in other domains, effectively integrating it into entity alignment models still necessitates further research and exploration [12,13].

To address these problems, we propose an Attribute-enhanced Relationship-Aware Neighborhood Matching Model with Dual Attention named ARNM-DAE2A for entity alignment. This model aims to fully exploit the correlations among entity attributes and accurately model entity relationships through the integration of attribute information and dual attention mechanism. The main contributions of this work are as below:

- We introduce a relationship-aware neighborhood matching model that incorporates a dual attention mechanism, further enhancing the structural learning capability of GCN.

- We introduce an innovative method for entity alignment that involves simultaneous learning of relationship structures and attribute details. These are embedded as preliminary matrix representations, followed by utilizing the relationship-aware neighborhood matching model for entity alignment.
- Experimental results on DBP15K and SRPRS datasets demonstrate that the proposed model achieves improved alignment accuracy compared to baselines.

## 2 Related work

### 2.1 Entity alignment based on GCN

Wang et al. [3] first introduced graph convolutional networks (GCN) into the task of entity alignment, marking the initiation of using GCN-based methods as a benchmark for model expansion and attracting wide-spread attention from researchers. Since then, the focus of research has gradually shifted from translation-based models like TransE to models based on GCN, igniting a research trend with GCN models at its core. Although the GCN-Align model proposed by Wang et al. applied GCN to entity alignment for the first time, the alignment effect was limited by only using the basic GCN. Cao et al. [14] incorporated GCN into entity alignment using a multi-channel graph neural network (MuGNN), which complements multi-channel approaches involving attention-based KG completion and cross-lingual attention-based entity pruning. Wu et al. [15] designed a relationship-aware dual-graph convolutional network (RDGCN), introducing the concept of bidirectional dual graphs and enhancing discrimination between different entity network structures by constraining these dual graphs. However, they neglected the dependence between attribute information utilization and relationship, and therefore the alignment effect was not further improved. Zhu et al. [16] devised relationship-aware neighborhood matching (RNM), using a relationship-aware method that incorporates neighborhood matching to refine GCN-based entity alignment. Zhu et al. designed RNM that used relational perception to correct GCN alignment, but also did not fully consider the aspect of attributes. Wu et al. [17] introduced a neighborhood matching network (NMN) designed to handle the heterogeneous neighborhoods within KGs. This approach estimated entity similarity by considering both topological structure and neighborhood similarities. Although the NMN and HGCN reflected the idea of neighborhood matching, they did not deeply optimize the combination of attributes and relationships. Additionally, Wu et al. [18] introduced hyperbolic graph convolutional networks (HGCN), using a high-speed gate mechanism to regulate the spread of noise within GCN structures, and leveraging entity representations for approximating relationship representations to optimize the goal of relationship alignment. Although the NMN and the HGCN reflected the idea of neighborhood matching, they did not deeply optimize the combination of attributes and relationships. GCN, as a neural network, can effectively learn the dependencies and connection rules in structural relationships, and extract structural information to provide strong support. However, most GCN-based entity alignment methods mainly focus on learning structural information, ignoring the importance of attribute features. The interactive learning between different information is not deep enough, and the fusion effect of attribute and structural information is low. In addition, it is difficult to balance the proportion of structural and attribute information in the model, which directly affects the alignment effect.

### 2.2 Entity alignment based on attribute information and attention mechanism

Sun et al. [19] introduced the AliNet that harnesses the local structures of multi-hop entities and enforces equivalent entity pairs to possess identical hidden states across each layer of the

graph attention network (GAT). However, AliNet relied too much on local structure and did not make good use of global context in-formation, which can easily lead to misalignment. Xin et al. [20] aggregated contextual information through a Transformer model and designed holistic reasoning based on embedding similarity, relationships, and entity functionality to evaluate alignment probabilities. Xin et al. used Transformer to collect context information, but ignored the correlation between structure and attributes, and could not express the complex features of entities well. Mao et al. [21] considered partitioning relationships into meta-relationships within a graph neural network framework, learning attention parameters among these meta-relationships and integrating them into entity representations, and ultimately training the model in a semi-supervised manner. Mao et al. learned relational attention under the GCN framework, but truncated the relationships into independent units, losing the order and continuity information. Furthermore, there were approaches relying on attention mechanisms, long short-term memory (LSTM), and the bidirectional encoder representations from transformers (BERT) model [22]. Attention mechanisms make up for the deficiencies of GCN in learning local relationships, and can effectively enhance the transfer of information related to the target. However, excessive reliance on local perspectives can also ignore the influence of global connectivity. When learning global contextual information, attention-based methods generally perform worse than GCN. In addition, how to master local and global information in the model and effectively combine them is a major problem that this method needs to address, and the current solution is not effective.

## 2.3 Entity alignment based on neighborhood matching and structural optimization

Wu et al. [15] utilized an attention mechanism to capture the interaction between the primary graph and the dual relation graph. Similar to Sun et al. [23] and Wang et al. [3], they utilized entity attribute information for neighborhood matching. However, the learning of entity relationship structures was not comprehensive enough, which could lead to the neglect of important entity information. Zhuang et al. [24] enhanced the modeling capability of different entity network structures by constraining the dual graph, indicating the potential effectiveness of this structural optimization approach in entity alignment tasks. Chami et al. [25] introduced a high-speed gate mechanism to control noise propagation in graph convolutional networks, optimizing approximate relationships. This structural optimization approach could be employed to improve relationship alignment performance. Zhu et al. [16] utilized neighborhood matching to enhance entity alignment, exploring useful information not only from neighboring nodes but also from connection relationships. Furthermore, iterative frameworks were designed to enhance structural optimization in a semi-supervised manner, fully harnessing the beneficial interactions between entity alignment and relationship alignment.

## 3 Problem definition

First, we define the KG as:

$$K = \{(E, R, V, A, T) | T \subset (E \times R \times E) \vee (E \times A \times V)\}, \tag{1}$$

where $E$, $R$, $V$, $A$ and $T$ are distinct components within the KG [26], with $E$ denoting entities, $R$ denoting relationships, $V$ denoting attributes, $A$ denoting attribute values, and $T$ denoting the set of knowledge triples, respectively.

Then, we define a relationship triple in the KG as:

$$T_R = \{(h, r, t) \mid h \in E, r \in R, t \in E, (h, r, t) \in T\}, \tag{2}$$

where $h$, $r$ and $t$ denote the head, relationship, and tail of a relationship triple, respectively.

Finally, we define an attribute triple as:

$$T_A = \{(h, a, v) \mid h \in E, a \in A, v \in V, (h, a, v) \in T\}, \tag{3}$$

where $h$, $a$ and $v$ denote the head, attribute, and attribute value of a attribute triple, respectively.

Given two KGs, denoted as $G_1 = (E_1, R_1, A_1, V_1, T_1)$ and $G_2 = (E_2, R_2, A_2, V_2, T_2)$, and a pre-aligned entity pairs denoted as $L = \{(e_1, e_2) \mid e_1 \in E_1, e_2 \in E_2, e_1 = e_2\}$, the essential task of entity alignment is to identify the remaining equivalent entity pairs. Where, $e_1$ denotes a specific instance in the entity set $E_1$ of the KG $G_1$, and similarly, $e_2$ denotes another instance. Fig 1 shows an example of cross-language entity alignment, where the left nodes are from the Chinese KG (labels are presented in Chinese), and the right nodes are from the English KG (labels are presented in English). Each node on the left is a direct translation of its corresponding node on the right. In the mathematical description, the symbol $E$ denotes the set of all entities in the two KGs, and this correspondence forms the basis of the interaction between structure and attribute information in the alignment process of this model. The objective of entity alignment is to identify relation pairs that have the same semantic meaning within the given two KGs.

## 4 Model framework

We design a relationship-aware matching model that combines attribute information and dual attention mechanism named ARNM-DAE2A, as shown in Fig 2. The model consists four modules: entity structure embedding module, attribute structure embedding module, joint alignment module, and relationship-aware neighborhood matching module. The entity structure embedding module enhances the capability of RDGCN to learn entity structure

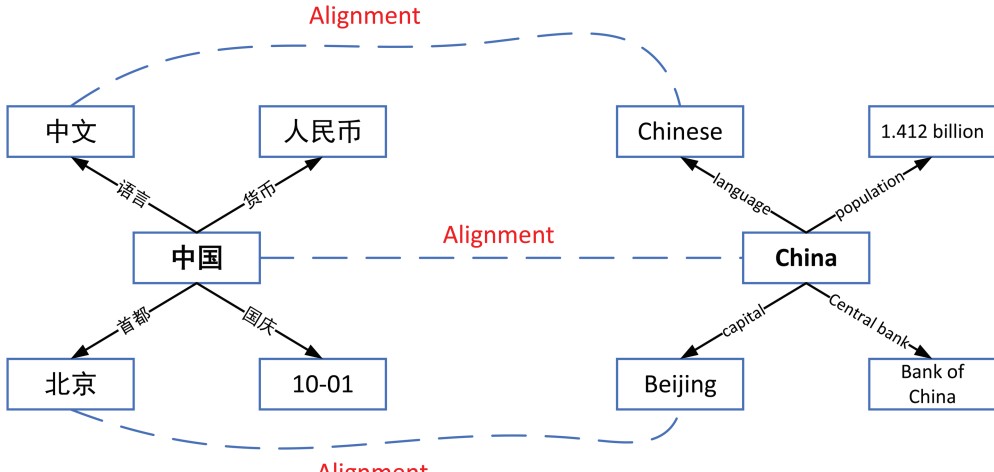

**Fig 1. An instance of cross-lingual entity alignment involving two KGs (Chinese KG and English KG).**

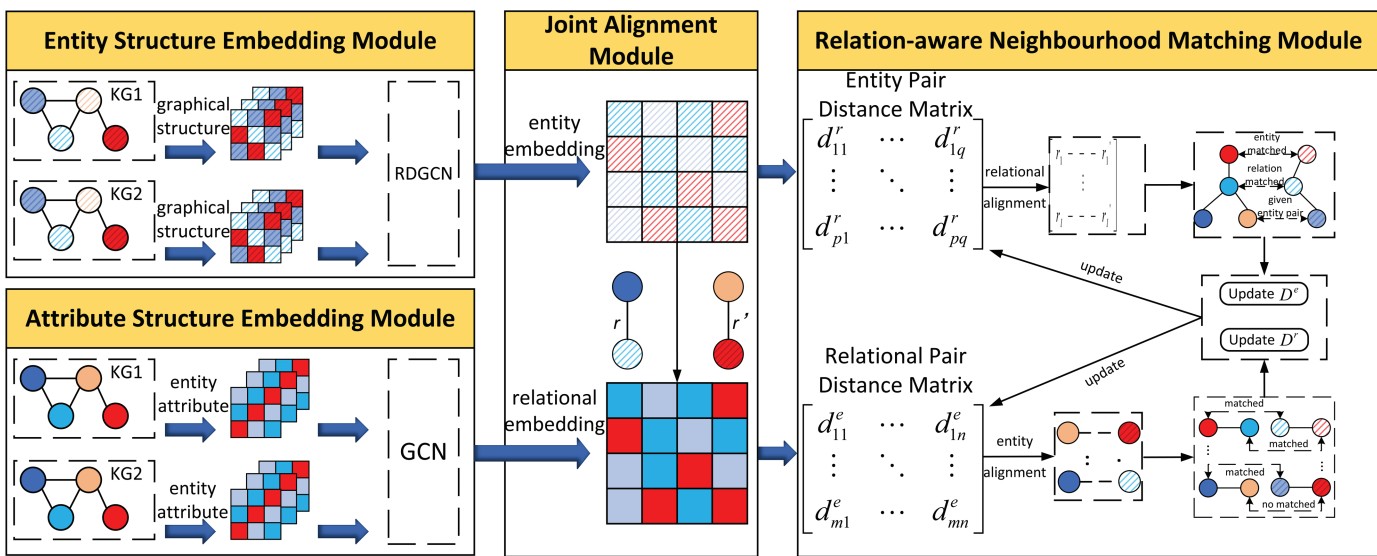

**Fig 2. Model framework.**

information by introducing a dual attention mechanism. The attribute structure embedding module employs GCN to learn entity attributes, providing additional feature representations. The joint alignment module combines relationship structure information and attribute information through weighted fusion, obtaining comprehensive entity representations. The relationship-aware neighborhood matching module corrects noise in GCN aggregation information by comparing neighborhood relations of entity pairs, thereby improving alignment accuracy.

## 4.1 Entity structure embedding module

In the traditional GCN, information transfer is primarily based on unidirectional neighborhood aggregation, which is susceptible to neighborhood noise and information asymmetry problems, resulting in less robust node representations. If information can be aggregated from both source and target nodes at the same time, i.e., bi-directional information interaction is realized, it helps improve the symmetry and stability of information transfer. Therefore, this module introduces the dual-attention mechanism to theoretically achieving noise suppression and key information reinforcement by adaptively assigning neighborhood weights, thereby improving the overall entity representation quality. To start, this module establishes a dual relationship graph derived from the original graph. It amplifies the interaction between the dual relationship graph and the original graph by incorporating attention mechanisms. Next, the module processes the nodes from the original graph through a GCN layer, enhanced with a high-speed neural network gating mechanism, to capture the structural information of neighboring nodes. Finally, the resulting entity representations from this module are utilized to assess the alignment of two entities.

**4.1.1 Constructing dual graph.** We denote the original graphs $G_1$ and $G_2$ as $\xi^e = (V^e, \varepsilon^e)$, where $V^e = E_1 \cup E_2$ denotes the set of nodes in the graph, and $\varepsilon^e = T_1 \cup T_2$ denotes the set of relations in the graph. For the two original KGs $\xi^e$, their dual relation graphs are denoted as $\xi^r = (V^r, \varepsilon^r)$ and defined as: 1) For each relation $r$ in $\xi^e$, there exists a vertex $v^r \in V^r$, resulting

in $V^r = R_1 \cap R_2$; 2) If relations $r_i$ and $r_j$ share entities or tail entities, edges $u_{ij}^r$ are created in $\xi^r$ to connect nodes $r_i$ and $r_j$.

Based on the above definition, we establish the edge weights within the dual relationship graph as:

$$\omega_{ij}^r = H(r_i, r_j) + T(r_i, r_j), \tag{4}$$

$$H(r_i, r_j) = \frac{H_i \cap H_j}{H_i \cup H_j}, \tag{5}$$

$$T(r_i, r_j) = \frac{T_i \cap T_j}{T_i \cup T_j}, \tag{6}$$

where $H_i$ denotes the set of head entities, and $T_i$ denotes the set of tail entities.

We use the graph attention network (GAT) [27] to iteratively obtain vertex representations of the dual relation graph and the original graph, where the attention mechanism helps facilitate interactions between the two graphs. The attention mechanism fosters interactions between these two graphs. Each bidirectional interaction comprises two layers: the dual attention layer and the original attention layer. This stacking of multiple interactions mutually reinforces both graphs.

(a) Dual Attention Layer

We represent the dual vertex representation matrix of the input KG as $X^r \in R^{m \times 2d}$, where each row denotes to a vertex within the relation graph $\xi^r$. We utilize the original node features $\ddot{x}^e$ generated by the original attention layer to calculate the dual attention scores:

$$\hat{x}_i^r = \sigma^r \left( \sum_{j \in N_i^r} \alpha_{ij}^r \bullet x_j^r \right), \tag{7}$$

$$\alpha_{ij}^r = \frac{\exp(\eta(\omega_{ij}^r \bullet a^r[c_i \parallel c_j]))}{\sum_{k \in N_i^r} \exp(\eta(\omega_{ik}^r \bullet a^r[c_i \parallel c_k]))}, \tag{8}$$

where $\hat{x}_i^r$ denotes the output representation at vertex $v_i^r$ in the dual graph, $x_j^r$ denotes the dual representation of vertex $v_j^r$, $N_i^r$ denotes the set of neighbor indices for dual vertex $v_i^r$, $\alpha_{ij}^r$ denotes the dual attention score, $a^r$ denotes the fully connected layer, $\sigma^r$ denotes the ReLU activation function, $\eta$ denotes the LeakyReLU activation function, $\parallel$ denotes the concatenation operation, and $c_i$ denotes the relation representation from the previous original attention layer.

By concatenating the average representations of head and tail entities in $\xi^e$, we obtain a relation representation $c_i$ obtained from the original attention layer:

$$c_i = \left[ \frac{\sum_{k \in H_i} \hat{x}_k^e}{|H_i|} \parallel \frac{\sum_{l \in T_i} \hat{x}_l^e}{|T_i|} \right], \tag{9}$$

where $\hat{x}_k^e$ denotes the output representation of the $k$-th head node of relationship $r_i$ obtained from an original attention layer, $\hat{x}_l^e$ denotes the output representation of the $l$-th tail node of $r_i$.

(b) Original Attention Layer

In this layer, when utilizing GAT for the original graph, we calculate the attention scores for the original graph using the dual vertex representations obtained from $\xi^r$, which are associated with the relations in the original KG $\xi^e$.

We denote the original vertex representation matrix of the input KG as $X^e \in R^{n \times d}$. For entity $e_q$ in the original KG $\xi^e$, its representation $\overline{\hat{x}_q^e}$ is shown as:

$$\hat{x}_q^e = \sigma^e \left( \sum_{t \in N_q^e} \alpha_{qt}^e \bullet x_t^e \right), \tag{10}$$

$$\alpha_{qt}^e = \frac{\exp(\eta(a^e(\widehat{x}_{qt}^r)))}{\sum_{k \in N_q^e} \exp(\eta(a^e(\widehat{x}_{qk}^r)))}, \tag{11}$$

where $\widehat{x}_{qt}^r$ denotes the dual representation of the relationship between entity $e_q$ and $e_t$ obtained in $\xi^r$, $N_q^e$ denotes the set of neighboring indices for entity $e_q$ in $\xi^r$, $\alpha_{qt}^e$ denotes the original attention score, $a^e$ denotes a fully connected layer, and $\sigma^e$ is an activation function.

In the model, the initial representation matrix $X_q^{e\text{-}init}$ for the original vertices can be initialized based on entity names, which offer crucial information for entity alignment [15]. Therefore, we retain the data by merging the output of the original attention layer with the initial representation:

$$\hat{x}_q^e = \beta_s \bullet \hat{x}_q^e + \hat{x}_q^{e\text{-}init}, \tag{12}$$

where $\hat{x}_q^e$ denotes the final representation of entity $e_q$ in the interactive module of the original KG, and $\beta_s$ denotes the weight parameters of the $s$-th layer of the original attention layer.

**4.1.2 Merge structural information.** Following numerous iterations involving the dual relation graph and the original graph, we can accumulate entity representations that are sensitive to relationships from the original graph. Next, we employ a dual-layer GCN with high-speed neural network gates [28] to further merge the structural information of neighbors.

At each GCN layer $l$, we take the entity representations $X^{(l)}$ as input:

$$X^{(l+1)} = \varsigma \left( \tilde{D}^{-\frac{1}{2}} \tilde{A} \tilde{D}^{-\frac{1}{2}} X^{(l)} W^{(l)} \right), \tag{13}$$

where $\tilde{A} = A + I$ denotes the adjacency matrix of the original graph $\xi^e$, $I$ denotes the identity matrix, $\varsigma$ denotes the activation function, $W^{(l)}$ denotes the weight for the $l$-th layer, and $D$ denotes the degree matrix. To enable bidirectional information flow, when constructing matrix $A$, we treat $\xi^e$ as an undirected graph.

**4.1.3 Relationship structure alignment.** Inspired by RDGCN[12], we gather the ultimate entity representations denoted as $\bar{X}$ from the output of the GCN layers, and the distance between two entities serves as the alignment score:

$$D(e_1, e_2) = \left\| \bar{x}_{e_1} - \bar{x}_{e_2} \right\|_{L_1}, \tag{14}$$

where $\bar{x}_{e_1}$ denotes the entity representation of entity $e_1$ in $G_1$, $\bar{x}_{e_2}$ denotes the entity representation of entity $e_2$ in $G_2$, and $D(e_1, e_2)$ denotes the distance between $e_1$ and $e_2$ in the same $L_1$ norm-regularized space.

## 4.2 Attribute structure embedding module

In real KGs, it is often difficult to rely on structural information alone to distinguish entities with similar connection patterns but subtle semantic differences. The attribute data of entities, such as names, descriptions, and other semantic labels, can provide complementary information that enhances the discriminative power of entity representations. Based on this, this module uses an independent GCN to model the entity attributes and vectorize the extracted

attribute features. Attribute information has a high degree of discrimination, which can distance structurally similar but semantically different entities in the embedding space, making it easier for the model to capture pairs of entities that are truly semantically related when aligned. With attribute-enhanced embedding, we hope to use this additional information to compensate for the limitations of single-structure embedding and further improve the overall discriminative power and alignment accuracy of the model. Therefore, in this module, we adopt the idea of graph convolutional alignment to separately analyze attributes. Unlike Wang et al. [3], we do not directly combine attribute information with GCN in joint training. Instead, we use a complete attribute matrix without considering its connection with the relationship structure matrix. We treat attributes as nodes, forming a KG where entities are connected by edges representing attributes. If an entity possesses a certain attribute, there is an edge connecting the entity to that attribute. Therefore, in the connection matrix of the GCN model, entities are only connected to attributes, and there are no connections between entities. The dimension of the connection matrix corresponds to the number of attributes. We initialize node vectors with random values as input for the first layer, and the convolutional process as:

$$H_a^{(l+1)} = \sigma(\hat{D}_a^{-\frac{1}{2}} \hat{A}_a \hat{D}_a^{-\frac{1}{2}} H_a^{(l)} W_a^{(l)}), \tag{15}$$

where $\hat{A}_a, H_a^{(0)} \in R^{N_a \times N_a}$, $N_a$ denotes the number of attributes, $a$ denotes the attribute structure embedding module, and $\sigma$ denotes the activation function.

The loss function for this module is defined as:

$$L_{ae} = \sum_{(e_i^a, e_j^a) \in S} \sum_{(e_i^a, e_j^a) \in S'} \max\{0, \gamma_a + f(e_i^a, e_j^a) - f(e_i^{a'}, e_j^{a'})\}, \tag{16}$$

where $\gamma_a > 0$ denotes the margin hyperparameter, $f(x, y) = \|x, y\|_1$, and $e^a$ denotes the entity-based embedding vector using attribute structure. We utilize stochastic gradient descent (SGD) [29] to minimize the aforementioned loss function.

## 4.3 Joint alignment module

After obtaining entity embedding based on both relationship structure and attribute structure, we compute the similarity between entities from these two aspects separately. Subsequently, we combine these similarities through weighted summation to derive the overall similarity between entities. The final entity similarity distance function is defined as:

$$D(e_i, v_j) = \varepsilon_{s+a} \frac{f(h_s(e_i), h_s(v_j))}{d_s} + \lambda_{s+a} \frac{f(h_a(e_i), h_a(v_j))}{d_a}, \tag{17}$$

where $f(x - y) = \|x - y\|_1$, $h_s(\bullet)$, and $h_s(\bullet)$ denote the embedding of relationship structure and attributes, respectively. $d_s$ and $d_a$ are the dimensions of the embedding for relationship structure and attributes, while $\varepsilon_{s+a}$ and $\lambda_{s+a}$ are hyperparameters that balance the importance of the two types of embedding.

## 4.4 Relationship-aware neighborhood matching module

**4.4.1 Entity embedding.** We use the inputs of GCN as entity embedding and define the representation form of entities as:

$$\tilde{X} = \left\{ \tilde{x}_1, \tilde{x}_2, \cdots, \tilde{x}_n \mid \tilde{x}_i \in R^{\tilde{d}} \right\}, \tag{18}$$

where $\tilde{d}$ denotes the dimension of entity embedding, $n$ denotes the number of entities. For an entity pair $\{(e_i, e_j') | e_i \in E_1, e_j \in E_2\}$, $(e_i, e_j)$ denotes the aligned entity pairs, and $(e_i, e_j')$ denotes the candidate entity pairs for alignment. We define the distance as $d(e_i, e_j') = \left\| \tilde{x}_{e_i} - \tilde{x}_{e_j'} \right\|_1$, where $\|\bullet\|_1$ denotes the vector norm measure. The smaller the distance $d(e_i, e_j')$ between entity pair $(e_i, e_j')$, the higher the probability that entity pair is aligned.

To embed entities from two KGs into a shared latent space, we employ seed alignment as training data and devise an edge-based loss function for entity alignment:

$$L_E = \sum_{(p,q) \in L} \sum_{(p',q') \in L'} \max\{0, d(p,q) - d(p',q') + \gamma\}, \tag{19}$$

where $L$ denotes the set of aligned seed entity pairs, $L'$ denotes a set of auxiliary alignments sampled from nearest neighbors, and $\gamma$ is a margin hyperparameter that separates positive and negative entity alignment pairs. Our loss function is designed with the assumption that the distance between aligned entity pairs should approach zero, whereas the distance between negative samples should be maximized [30].

**4.4.2 Relationships embedding.** We employ the information linking entities and the embedding of head and tail entities acquired through GCN to represent relationships within the KGs:

$$r = concat[g_r^h, g_r^l], \tag{20}$$

where $r \in R^{2d}$ denotes the embedding of relationship $r \in R_1 \cup R_2$, $concat[\bullet]$ denotes the concatenation operation, $g_r^h$ and $g_r^l$ respectively signify the average embedding of all distinct head and tail entities of relationship $r$.

Furthermore, to delve deeper into the translational insights stemming from ternary relationships, we employ a regularize similar to the TransE model [31]:

$$\Omega_R = \sum_{(h,r,t) \in T_1 \cup T_2} \|h + W_R r - t\|_1, \tag{21}$$

where $T_1$ and $T_2$ denote the triple sets of the given KGs $G_1$ and $G_2$, respectively. $W_R \in R^{\tilde{d} \times 2\tilde{d}}$ denotes the transformation matrix from the latent relationship space to the latent entity space, which is the model parameter to be learned.

To simultaneously acquire embedding for entities and relationships, we optimize the objective function following the pretraining of entity embedding:

$$L = L_E + \lambda \Omega_R, \tag{22}$$

where $\lambda$ is a weighting coefficient used to balance the loss of entity alignment and the loss of regularize when considering the relation prior.

**4.4.3 Relationship-aware neighbor matching for entity pairs.** For each candidate entity pair $\{(e_i, e_j') | e_i \in E_1, e_j \in E_2\}$, in addition to comparing their one-hop neighbor entities in pairs, we also consider comparisons between connectivity relations [13]. Assuming that $N_{e_i}$ is the set of first-order neighboring entities of $e_i$ in $G_1$ and $N_{e_i'}$ is the set of first-order neighboring entities of $e_j'$ in $G_2$, for neighborhood matching between $e_i$ and $e_j'$, we compare $C_{ij}^e = \{(n_1, n_2), (r_1, r_2) | n_1 \in N_{e_i}, n_2 \in N_{e_j'}\}$. After that, we focus on matching neighbours with matching relations. In addition, the mapping property of the connectivity relation is also important for entity alignment. Therefore, for each matching case in the matching set $M_{ij}^e$, we

compute the alignment probability based on $r_1$, $r_2$, $n_1$ and $n_2$:

$$p(r_1, r_2, n_1, n_2) = P(r_1, n_1) \bullet P(r_2, n_2), \quad (23)$$

where $P(r_1, n_1) = \frac{1}{|\{e|(e,r_1,n_1) \in T_1\}|}$, $P(r_2, n_2) = \frac{1}{|\{e|(e,r_2,n_2) \in T_2\}|}$, they denote the mapping probabilities of correspondence and neighboring entities, respectively. The distance between two entities is updated as:

$$d_{ij}^e = \left\| \tilde{x}_{e_i} - \tilde{x}_{e_i'} \right\|_1 - \lambda_e \bullet \frac{\sum_{M_{ij}^e} P(r_1, r_2, n_1, n_2)}{|N_{e_i}| + \left| N_{e_j'} \right|}, \quad (24)$$

where $\lambda_e$ denotes a hyperparameter that regulates the balance between embedding distance and matching score. A higher matching score increases the likelihood of aligning candidate entity pairs.

**4.4.4 Entity-aware relationship matching.** For relation $r$, we denote the set of associated entity pairs as $S_r = \{(h,t) | (h,r,t) \in T\}$, where $T$ denotes the set of triples in the given KG. Given a candidate pair of relations $(r_i, r_j')$, $r_1$ from $G_1$ and $r_j'$ from $G_2$. First, we form corresponding sets of entity pairs $S_{r_i}$ and $S_{r_j}$. Then, we compare all entity pairs in $C_{ij}' = \{(h_1, h_2), (t_1, t_2) | (h_1, t_1) \in S_{r_i}, (h_2, t_2) \in S_{r_i'}\}$ and define the subset $M_{ij}^e$ with the matching set $C_{ij}^r$. The distance between the updated relationship pairs is:

$$d_{ij}^r = \left\| r_i - r_j' \right\|_1 - \lambda_r \bullet \frac{\left| M_{ij}^r \right|}{|S_{r_i}| + \left| S_{r_j'} \right|}. \quad (25)$$

## 4.5 Complexity analysis

GCN is responsible for learning the structured representation of entities, and the main computational effort is from neighborhood feature aggregation and feature transformation. Assuming that a KG contains $N$ entities and $E$ edges with embedding dimension $d$ and GCN adopts $L$ layers, the complexity of neighborhood feature aggregation is $O(Ed)$, the complexity of feature transformation is $O(Nd^2)$, and the overall computational complexity is $O(L(Ed + Nd^2))$.

The dual attention mechanism is used to optimize the GCN propagation process by introducing additional attention weight calculation and weighted neighbor aggregation. Assuming that the average number of neighbors of each entity is $k$, the complexity of attention computation is $O(Nkd)$, the complexity of weighted aggregation is also $O(Nkd)$, and the overall complexity is $O(LNkd)$.

The joint alignment module calculates the entity similarity by fusing structural and attribute information, and the computational complexity is determined by embedding fusion and similarity calculation. Here, the computational complexity of the fusion process is $O(Nd)$, the complexity of similarity computation is $O(Md)$ based on the number of entity pairs $M$, and the overall complexity is $O(Nd + Md)$.

The relationship-aware neighborhood matching module further adjusts the entity alignment results, and its computational complexity is primarily determined by the neighborhood matching computation. Here, the complexity of the neighborhood matching operation for each entity is $O(Nkd)$.

In summary, the overall computational complexity of ARNM-DAE2A is $O(L(Ed + Nd^2)) + O(LNkd) + O(Nd + Md) + O(Nkd)$. The GCN computation grows linearly with the number of layers $L$, the neighborhood matching computation depends on the number of neighbors $k$, and the cost of the alignment computation is determined by the number of entity pairs $M$. The GCN computation is also a good example of GCN computation. It is worth mentioning that the computational cost may rise significantly in the case of dense graphs ($k \neq N$) or large-scale entity comparisons ($M \neq N^2$).

## 5 Experimental Analysis

### 5.1 Datasets

We conducted simulation experiments using the DBP15K dataset [32], and comprehensive datasets details are presented in Table 1. DBP15K is a large-scale KG of a comprehensive encyclopaedia, containing rich semantics in different languages, with links across different languages, making it suitable for cross-lingual datasets. Since the DBP15K dataset is too dense and the degree distribution is very different from the real-world data, we also chose two cross-linguistic subsets from the relatively sparse SRPRS dataset [33], and comprehensive dataset details are presented in Table 2. For comparison with previous works, we used the same training/testing split with previous works [15], 30% for training and 70% for testing.

### 5.2 Experimental setup

**5.2.1 Experimental platform.** We conducted experiments on a server with the specifications of Intel 4210R/2x Tesla V100-32G GPUs, an 8-core CPU, and 40GB of RAM. The model was implemented using the TensorFlow framework.

**5.2.2 Evaluation metrics.** We use Hits@k and mean reciprocal rank (MRR), which are commonly used in KG entity alignment tasks, as evaluation metrics to evaluate all methods as:

$$Hits@k = \frac{1}{N} \sum_{i=1}^{N} Z(rank_i \leq k), \tag{26}$$

**Table 1. Information of the DBP15K.**

| DBP15K | | Entities | Attributes | Relationship triples |
|---|---|---|---|---|
| ZH-EN | Chinese | 66496 | 8113 | 153929 |
| | English | 98125 | 7173 | 2376474 |
| JA-EN | Japanese | 65744 | 5882 | 164373 |
| | English | 95680 | 6066 | 233319 |
| FR-EN | French | 66858 | 4547 | 192191 |
| | English | 105889 | 6422 | 278590 |

**Table 2. Information of the SRPRS.**

| SRPRS | | Entities | Attributes | Relationship triples |
|---|---|---|---|---|
| FR-EN | French | 15000 | 177 | 33532 |
| | English | 15000 | 221 | 36508 |
| DE-EN | German | 15000 | 120 | 37377 |
| | English | 15000 | 222 | 38363 |

$$MRR = \frac{1}{N} \sum_{i=1}^{N} \frac{1}{rank_i}, \qquad (27)$$

where $N$ is the number of entities, $rank_i$ is the ranking of entities aligned with the $i$-th entity, and $z(\bullet)$ is an indicator function (the value of the function is 1 if the condition is true and 0 otherwise). For each source entity, entities in the other KGs are ranked in descending order based on their similarity to the source entity. $Hits@k$ measures the alignment accuracy, indicating the percentage of correctly aligned entities among those similar to the source entity. Meanwhile, MRR provides complementary insights into the entity alignment results. The higher the values of Hits@k and MRR, the better the performance of the model.

**5.2.3 Parameter setting.** In the entity structure embedding module, according to [15], the hidden dimensions for both the dual attention layer and the original attention layer are set to 300. The hidden dimensions for all layers in the GCN are also set to 300. The learning rate is set to 0.001, and negative entity pairs are generated every 10 epochs of training, accumulating a total of 600 epochs.

For the attribute structure embedding module, we employ the SGD optimization algorithm for 2000 iterations of model updating. According to [15], the dimension $d_a$ is set to 300, and the output, input, and final dimensions of the first and second layers of GCN are kept equal.

In the joint alignment module, according to [3], the parameters $\varepsilon_{s+a}$ and $\lambda_{s+a}$ in (17) are set to 0.9 and 0.1, respectively.

In the relationship-aware neighborhood matching module, according to [3], a 2-layer GCN is utilized to learn entity embedding. The hidden dimensions for both structure and attribute are set to 300, and the learning rate is set to 0.001. According to [34], seed alignment is set at a ratio of 30%. The boundary $\gamma$ is set to 1, the learning rate $\lambda$ is set to 0.001, $\lambda_e$ is set to 10, and $\lambda_r$ is set to 200. The nearest 100 entities and 20 relations are selected as candidates for matching. Each positive sample is paired with 125 negative samples, and a maximum iteration count $T$ is set to 4. Optimization using (19) is performed for 50 epochs, followed by joint embedding training using (22) for an additional 10 epochs.

## 5.3 Experimental results and analysis

**5.3.1 Comparative experiments.** For evaluating our proposed ARNM-DAE2A model, we compare it with the state-of-art methods: JAPE [23], GCN-Align [3], RDGCN [15], CTEA [35], HMAN [36], Dual-AMN [37], and MSNEA [38].

The comparative experimental results on DBP5K are shown in Table 3, where the results are presented as percentages rounded to two decimal places. Bold values indicate the best performance.

**Table 3. Result of comparative experiments on DBP15K.**

| Model | ZH-EN | | | | JA-EN | | | | FR-EN | | | |
|---|---|---|---|---|---|---|---|---|---|---|---|---|
| | Hits@1 | Hits@10 | Hits@50 | MRR | Hits@1 | Hits@10 | Hits@50 | MRR | Hits@1 | Hits@10 | Hits@50 | MRR |
| JAPE | 41.18 | 74.46 | 88.90 | 49.00 | 36.25 | 68.50 | 85.35 | 47.60 | 32.39 | 66.68 | 83.19 | 43.00 |
| GCN-Align | 41.25 | 74.38 | 86.23 | 54.90 | 39.91 | 74.46 | 86.10 | 54.60 | 37.29 | 74.49 | 86.73 | 53.20 |
| RDGCN | 70.75 | 84.55 | N/A | 74.90 | 76.74 | 89.54 | N/A | 81.20 | 88.64 | 95.72 | N/A | 90.80 |
| CTEA | N/A | 90.50 | N/A | 70.70 | N/A | **91.40** | N/A | 70.40 | N/A | 92.30 | N/A | 71.50 |
| HMAN | 56.20 | 85.10 | 93.40 | N/A | 56.70 | 86.90 | **94.50** | N/A | 54.00 | 86.70 | 95.10 | N/A |
| Dual-AMN | 83.13 | 91.43 | 95.44 | 88.12 | **88.23** | 86.69 | 90.33 | 90.89 | 89.13 | 97.23 | 97.26 | 92.33 |
| MSNEA | 83.83 | 92.14 | 96.12 | 88.23 | 87.66 | 89.33 | 92.71 | 91.22 | 91.31 | 98.24 | 97.97 | 93.89 |
| ARNM-DAE2A | **83.91** | **92.38** | **96.46** | **88.34** | 87.71 | 90.76 | 93.86 | **91.37** | **91.50** | **98.32** | **98.96** | **95.60** |

JAPE systematically introduces attribute information for the first time in a representation learning framework, incorporating both the structural and attribute information of entities into the embedding space to alleviate the heterogeneity problem in cross-lingual entity alignment. Attributes can provide additional discriminative features to help distinguish structurally similar but semantically different entities. However, early representation learning methods generally have poor accuracy benchmarks and are immature in fusing multiple modal information, resulting in poor performance of JAPE. For instance, on the ZH-EN dataset, the Hits@1 is only 41.18%.

GCN-Align introduces GCN into the entity alignment task for the first time and learns it jointly with attribute information. GCN can theoretically be regarded as a local filtering performed on the graph structure to obtain the contextual relationship of entities in the graph spectrum through neighborhood aggregation. Therefore, there are more GCN-based entity alignment methods in the current entity alignment tasks [22]. However, GCN-Align uses only the most basic GCN and is relatively inadequate in handling noise and heterogeneous structures, e.g., Hits@1 is only 41.25% on ZH-EN. This reflects that simply using GCNs for entity alignment cannot yet fully mitigate the complex heterogeneity in cross-lingual knowledge graphs.

CTEA and HMAN, on the other hand, take the GCN as the base model and utilize more auxiliary information to further improve the performance. CTEA adopts GCN and TransE to process entity alignment in parallel, which theoretically integrates two different representations of structural and translation models, and the Hits@10 on ZH-EN can reach 90.50%. However, due to the lack of deeper coupling or improvement of the GCN and TransE, the improvement of Hits@1 is still limited. HMAN attempts to mine richer semantic features from the context by introducing a BERT variant model, but the high time complexity and inconvenient use of BERT results in only 56.20% of Hits@1 on ZH-EN, which indicates that the task still needs to strike a balance when introducing large language models.

Dual-AMN further emphasizes the role of dual attention and incorporates difficult sample mining to accelerate the alignment process. Theoretically, dual-attention can focus on both the local neighborhood and the global structure to reduce the interference of noisy entities, while difficult sample mining can strengthen the model's learning ability near the discriminative boundary. Experiments show that the method can significantly improve the alignment speed while maintaining high alignment accuracy, indicating that the combination of multi-angle attention and difficult sample mining has high utility in the alignment task.

MSNEA emphasizes characterizing entities from a Multi-Scale perspective, fusing different levels of graph structure and semantic information. According to the theory of multimodal feature fusion, information from multiple scales tends to provide a more comprehensive contextual semantics, thus helping models to better distinguish similar entities in the alignment task. The method also achieves excellent performance on cross-language datasets, suggesting that multi-scale representation can effectively alleviate the semantic divide across languages or modalities.

Consistently across multiple datasets, our proposed ARNM-DAE2A model outperforms existing baseline models on DBP15K, establishing itself as the state-of-the-art solution. Specifically, the ARNM-DAE2A model leverages an attribute information module to learn entity attribute features, optimizes structural learning with RDGCN, combines structural and attribute information for multi-dimensional entity representation, and utilizes a relationship-aware matching mechanism to correct noise introduced by GCN's aggregation of neighbor information. Finally, the model employs a cyclic iterative learning framework that positively influences and mutually enhances entity alignment and relation alignment tasks. These

innovations enrich entity representations, enhance structural learning capabilities, effectively integrate heterogeneous information sources, and improve the model's resilience to disturbances.

To more intuitively demonstrate the performance of the ARNM-DAE2A model, we compared it with multiple benchmarks on the DBP15K dataset using Hits@1 to Hits@50 with a step size of 10. We selected JAPE, GCN-Align, and RD-GCN as the comparison models, as shown in Fig 3. It can be seen that the Hits@K values of our proposed ARNM-DAE2A model are higher than those of other models, achieving the highest score on both the JA-EN and FR-EN datasets. The Hits@K value of the GCN-Align model approaches our ARNM-DAE2A model after K is set to 20, indicating that alleviating the heterogeneity of the entity neighborhood structure is beneficial to entity alignment. However, the Hits@1 of the GCN-Align model is significantly lower than that of the ARNM-DAE2A model, indicating that the ARNM-DAE2A model has better alignment performance.

The comparative experimental results on SRPRS are shown in Table 4, where the results are presented as percentages rounded to two decimal places. Bold values indicate the best performance. Compared with the results on DBP15K, the overall performance trend of the state-of-the-art methods on SRPRS remains largely the same, indicating that their strengths and weaknesses do not fundamentally change under different map sizes and densities. However, ARNM-DAE2A shows a more obvious relative advantage in this setting, even exceeding its achievement on DBP15K. First, the attribute structure embedding module effectively compensates for the lack of neighborhood information, and the complementary nature of the entity attributes becomes more prominent when the graph structure is sparse. Second, the dual-attention mechanism can perform bi-directional aggregation between the source and target entities, and is more resistant to noise and missing connections. Third, the relationship-aware neighborhood matching by iteratively comparing pairing and correcting alignment results is particularly suitable for reducing error propagation in sparse graphs. Taken together, all these mechanisms can play a more significant compensatory role in sparse environments, making the alignment effect of ARNM-DAE2A even exceed its achievement on denser datasets.

Finally, we compare the average training time of each method on the two datasets, as shown in Table 5, where the results are presented as hours rounded to two decimal places. JAPE employs only direct representation learning, characterized by smaller parameter sizes and a concise computational process, resulting in a shorter training time. GCN-Align introduces a GCN to enhance entity representations through neighborhood aggregation, which increases the computational effort. RDGCN further incorporates relational graph modeling, enabling the model to capture complex graph structure information while significantly increasing computational complexity. CTEA and HMAN utilize TransE and BERT as auxiliary modules, respectively, to enhance representations. However, these auxiliary modules inevitably add to the computational burden. In contrast, the proposed ARNM-DAE2A fuses both structural and attribute information within the RDGCN framework and employs a dual-attention mechanism combined with relation-aware matching to correct for noise. Although this design results in a higher computational complexity, we believe that the additional training cost is acceptable, as it significantly improves both the accuracy of entity alignment and the robustness of the model, thereby demonstrating a substantial advantage in practical applications.

**5.3.2 Ablation experiments.** For further evaluation of our proposed ARNM-DAE2A model, we conducted ablation experiments on various modules within each model. We performed ablations by considering the model with only the matching module (MM), the matching module combined with structural embedding (MM+SE), and the matching module with

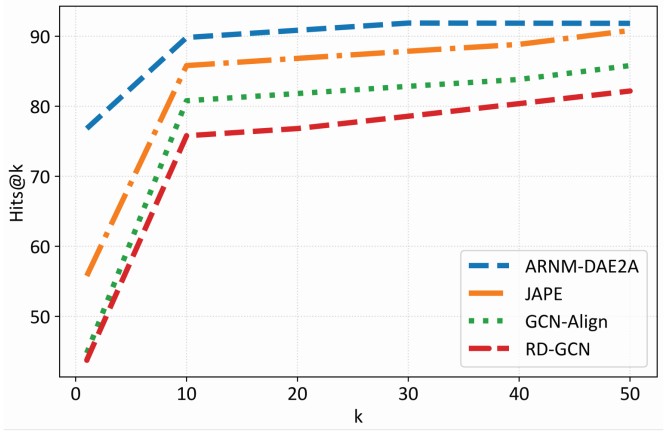

(A) ZH-EN.

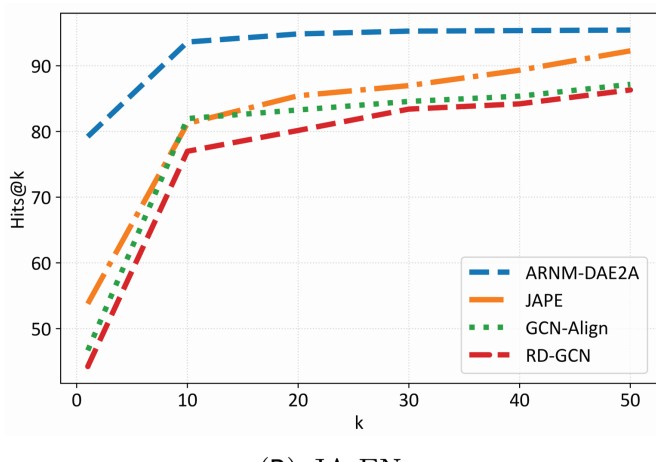

(B) JA-EN.

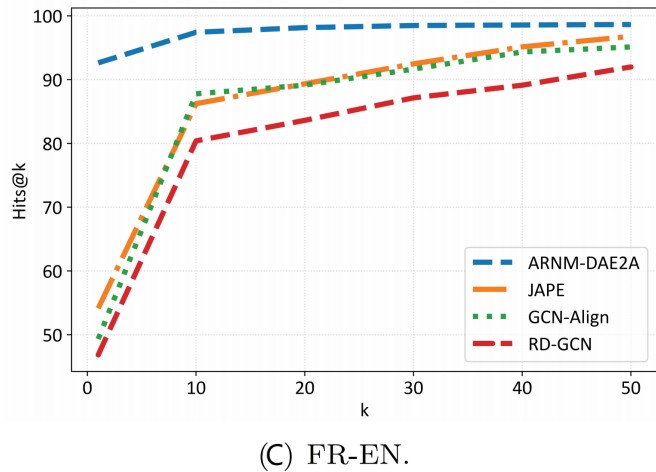

(C) FR-EN.

**Fig 3. Comparison of Hits@K results.** (**A**) ZH-EN. (**B**) JA-EN. (**C**) FR-EN.

**Table 4. Result of comparative experiments on SRPRS.**

| Model | FR-EN | | | | DE-EN | | | |
|---|---|---|---|---|---|---|---|---|
| | Hits@1 | Hits@10 | Hits@50 | MRR | Hits@1 | Hits@10 | Hits@50 | MRR |
| JAPE | 33.89 | 62.33 | 67.34 | 35.33 | 45.63 | 62.33 | 65.33 | 53.79 |
| GCN-Align | 24.36 | 52.31 | 66.32 | 34.01 | 38.51 | 60.03 | 71.33 | 46.04 |
| RDGCN | 67.22 | 76.73 | 81.45 | 71.01 | 77.91 | 88.62 | 90.33 | 82.01 |
| CTEA | 55.65 | 69.99 | 79.46 | 75.33 | 71.36 | 78.46 | 81.03 | 84.09 |
| HMAN | 40.11 | 70.51 | 81.94 | 50.03 | 52.81 | 77.83 | 83.33 | 62.00 |
| Daul-AMN | 80.23 | 83.22 | 90.33 | 85.13 | 89.13 | 92.21 | 93.33 | 92.30 |
| MSNEA | 86.22 | 87.33 | 91.46 | 89.77 | 90.46 | 93.13 | 94.26 | 93.33 |
| ARMN-DAE2A | **89.37** | **90.23** | **92.33** | **91.64** | **90.98** | **93.56** | **95.37** | **94.58** |

**Table 5. Results of training time.**

| JAPE | GCN-Align | RDGCN | CTEA | HMAN | Dual-AMN | MSNEA | ARMN-DAE2A |
|---|---|---|---|---|---|---|---|
| 3.62 | 4.53 | 6.31 | 5.54 | 7.32 | 7.51 | 7.31 | 7.96 |

both structural embedding and attribute embedding (MM+AE), as shown in Table 6 and Table 7.

Using only the MM for entity alignment yields poorer results compared to the baseline. This indicates that when relying solely on neighborhood matching, the entities themselves lack sufficient semantic representations, and the matching results are difficult to stabilize once the neighborhood information is limited or noisy. The effect of the matching module relies on the entity representation of the external input, which can constrain the upper limit of the effect of the matching module if the representation is limited. Meanwhile, the matching module only relies on the local neighbor information of the entities, which is prone to noise interference and lacks a global structural perspective, making it difficult to model symmetric relationships of entities and unable to play the role of relationship constraints in the KG. Comparing DBP15K and SRPRS, this trend is more obvious in both dense and sparse environments, indicating that the bottleneck of neighborhood matching mainly comes from the lack of semantic features of the entities. Lacking deep learning of the intrinsic structure or

**Table 6. Result of ablation experiments on DBP15K.**

| Model | ZH-EN | | | | JA-EN | | | | FR-EN | | | |
|---|---|---|---|---|---|---|---|---|---|---|---|---|
| | Hits@1 | Hits@10 | Hits@50 | MRR | Hits@1 | Hits@10 | Hits@50 | MRR | Hits@1 | Hits@10 | Hits@50 | MRR |
| MM | 83.57 | 91.73 | 93.00 | 87.20 | 86.95 | 94.39 | 95.53 | 89.70 | 90.38 | 97.96 | 98.44 | 95.33 |
| MM+SE | 83.53 | 91.84 | 93.40 | 87.20 | 86.90 | 94.59 | **96.10** | 89.50 | 91.41 | 96.34 | 97.24 | 93.30 |
| MM+AE | **84.91** | 92.17 | **96.51** | **89.70** | 87.67 | **95.76** | 95.19 | 90.70 | 91.45 | 97.30 | 98.46 | 95.33 |
| ARNM-DAE2A | 83.91 | **92.38** | 96.46 | 88.34 | **87.71** | 90.76 | 93.86 | **91.37** | **91.50** | **98.32** | **98.96** | **95.60** |

**Table 7. Result of ablation experiments on SRPRS.**

| Model | FR-EN | | | | DE-EN | | | |
|---|---|---|---|---|---|---|---|---|
| | Hits@1 | Hits@10 | Hits@50 | MRR | Hits@1 | Hits@10 | Hits@50 | MRR |
| MM | 84.24 | 88.73 | 89.92 | 88.68 | 89.93 | 92.56 | 93.86 | 93.62 |
| MM+SE | 86.97 | 89.99 | 90.37 | 87.63 | 88.64 | 93.31 | 92.56 | 92.56 |
| MM+AE | 88.31 | 89.76 | 91.86 | 90.30 | **91.30** | 92.99 | 94.22 | 93.47 |
| ARMN-DAE2A | **89.37** | **90.23** | **92.33** | **91.64** | 90.98 | **93.56** | **95.37** | **94.58** |

attributes of entities, neighborhood matching can only identify local relationships to a certain extent, but cannot globally grasp the semantic similarity between entities.

When the MM+SE for early-stage structural learning, some metrics show a relatively large improvement, suggesting that the structural representation brought by the GCN helps to make more accurate judgments in later matching. However, in some scenarios (e.g., ZH-EN of DBP15K), Hits@1 may show a small decrease, indicating that early GCN learning may instead bring some errors into later matching if it fails to converge sufficiently when there is more noise or higher sparsity. Nonetheless, overall, the SE module improves most of the metrics more than this cost, especially in sparse environments such as SRPRS, where the effect of complementing neighborhood gaps through structural information is more significant.

MM+AE yields impressive results, occasionally even surpassing the overall model's performance. This is because that attribute feature can in some cases provide very different discriminative information from the graph structure, allowing the matching module to be more targeted in correcting initial misalignments. However, Hits@10, Hits@50, or MRR do not necessarily outperform the full model at the same time, suggesting that relying on attribute features alone may also be a bottleneck in a wider range of entity retrievals. Once the attributes are inadequate or noisy, the advantage of the AE module is diminished and cannot replace the value of structural embedding for global information capture.

**5.3.3 Hyperparameter experiments.** To validate the impact of various hyperparameters in our proposed ARNM-DAE2A model on the experimental results, we conducted experiments with important model hyperparameters, taking Hits@1 as an example for visualization and analysis.

Firstly, we tested the entity dimension, as shown in Fig 4. When the dimension is between 0 and 1250, Hits@1 alignment results steadily increase. However, after reaching 1250 dimensions, the improvement in Hits@1 significantly diminishes. This could be due to the fact that as the dimensionality increases, the probability of similarity between two randomly generated high-dimensional vectors exponentially decreases. Therefore, increasing the dimensionality beyond 1250 has a diminishing impact on Hits@1, with a noticeable marginal effect.

Furthermore, we conducted tests on the number of GCN layers in the entity structure embedding module, as shown in Fig 5. When the number of layers is between 1 and 4, the Hits@1 results for alignment steadily improve. However, after increasing to 4 layers, Hits@1 not only stops improving but also declines. On one hand, this could be due to overfitting as

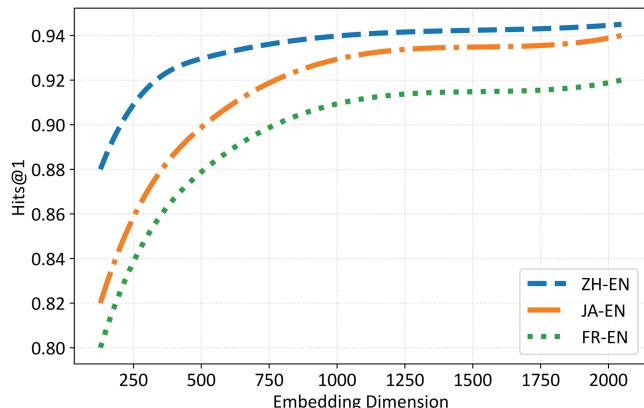

**Fig 4. Effect of different vector dimensions.**

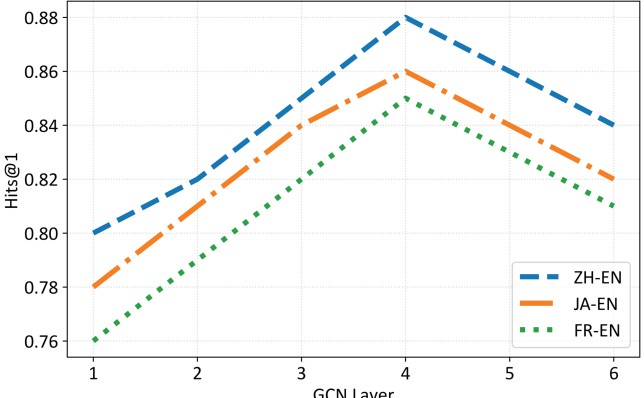

**Fig 5. Effect of different GCN layer.**

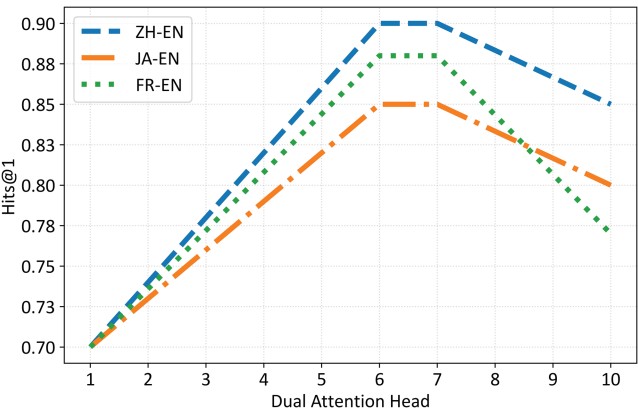

**Fig 6. Effect of different dual attention head.**

multiple layers of neighborhood aggregation make entity nodes' representations too similar, leading to the model's inability to generalize well to the test data and resulting in decreased performance. On the other hand, the diminishing returns might be attributed to the cumulative errors introduced by the chained matrix operations in multi-layer GCN, which could degrade the model's effectiveness.

Finally, we conducted tests on different numbers of dual attention heads, as illustrated in Fig 6. With an increase in the number of dual attention heads from 1 to 6, the model's representation capacity gradually improved, allowing it to better capture the relational information among entities, thereby enhancing alignment accuracy. However, when the number exceeded 7 heads, the model might begin to overfit the training data, leading to a decrease in performance on the test data. This could be attributed to the model's excessive sensitivity to the training data, making it challenging to generalize to other datasets or unseen entity pairs.

**5.3.4 Robustness experiments.** As shown in Fig 7A, embedding dimension in the range of 200 to 700 , the five subsets show a gradual upward trend and stabilize at 500 to 600 dimensions. This suggests that as the embedding dimension increases, the model can learn richer entity features, thus improving the alignment accuracy. However, when the dimension exceeds a certain threshold, the performance gain gradually diminishes, indicating that the

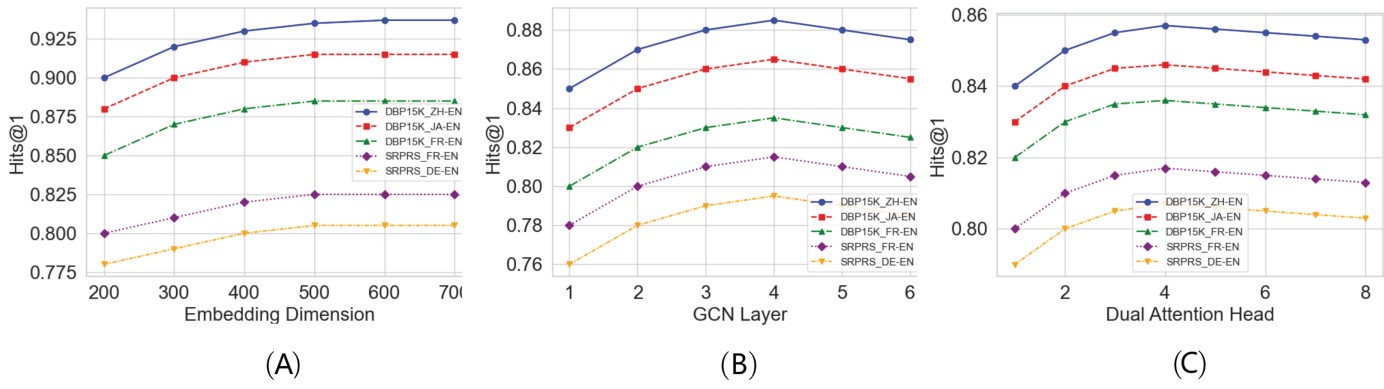

**Fig 7. Robustness with different hyperparameters.**

model is more robust to changes in the embedding dimension in the range of 500-600, and the performance degradation is limited even if the parameters slightly deviate from the optimal values. As shown in Fig 7B, with respect to the number of GCN layer, the result shows that as the number of GCN layers is increased from 1 to 4, the Hits@1 of the model is significantly improved in both datasets, which suggests that the increase in the number of layers can help to capture deeper graph structure information. However, the model performance slightly decreases when the number of layers exceeds 4, probably due to the effect of the over-smoothing phenomenon, which makes the node representations indistinguishable. This result shows that the model is more sensitive to the number of GCN layers, but within a reasonable range, the appropriate number of layers can effectively improve the model performance, while too many layers may lead to excessive aggregation of information, which may affect the alignment accuracy. As shown in Fig 7C, the result shows that in the range of 1 to 8, all the five subsets under perform best in the interval of 3 to 5 heads, followed by a slight decrease in performance as the number of dual attention heads increases. This suggests that an appropriate amount of dual attention heads can help the model capture multi-angle information more comprehensively and enhance entity matching. However, too few attention heads can limit the model's learning ability, while too many heads may introduce additional noise or lead to a waste of computational resources. Therefore, although the model is robust to parameter variations in the range of 3 to 5 heads, it still needs to be tuned within a reasonable interval to ensure optimal performance.

Referring to [39], we test the robustness of the model by artificially introducing different degrees of textual noise, where textual noise includes entity errors, missing words, and additional characters. As shown in Fig 8, the alignment accuracy of each dataset tends to decrease as the proportion of noise increases, but the magnitude of the decrease is significantly different. In the DBP15K subsets, Hits@1 decreases more gently, which indicates that the model can rely on sufficient structural and attribute information to mitigate the negative impact of noise in the case of richer information. In the SRPRS subsets, the decrease in Hits@1 is more pronounced due to the sparser graphs and less redundant information. This suggests that textual perturbations interfere more with entity alignment in noisy scenarios with sparse data. Overall, the experimental results illustrate that the proposed ARNM-DAE2A is robust under low to moderate noise levels, but there is still room for improvement under high noise and sparse data conditions, and further optimization of the noise processing and information fusion strategies is needed.

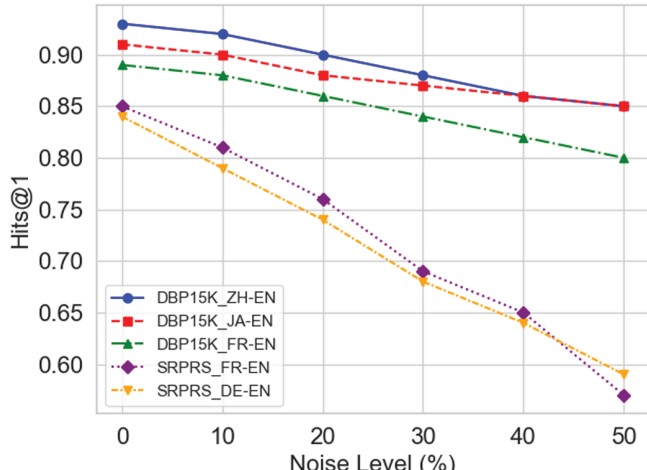

**Fig 8. Robustness with textual noise.**

**5.3.5 Case study.** We take Fig 1 as an illustrative example of a case study of the workflow of the dual attention mechanism. While the traditional one-way attention mechanism only passes information from entities to their neighbors, our dual attention mechanism calculates two sets of attention weights simultaneously. On the one hand, in the Chinese KG, by calculating the source attention from a Chinese-side entity to its neighboring nodes, the importance of each neighbor to the representation of that entity can be quantified; on the other hand, by calculating the target attention from the neighboring nodes back to that Chinese-side entity, the feedback contribution of the neighbors to the representation of the entity is reflected. Similarly, in the English KG, the exact bidirectional computation is performed for the corresponding English-side entity and its neighbor nodes. Subsequently, these two attention weights are fused to form a more comprehensive and robust entity representation. Based on this fused representation, the alignment process shown in Fig 1 can accurately capture the semantic consistency between the two sides of the KG in terms of languages, population, capital, and central bank, significantly improving the accuracy of entity alignment and the model's noise immunity.

# 6 Potential applications

KG entity alignment plays a crucial role in various real-world applications where integrating heterogeneous data sources is necessary. The proposed ARNM-DAE2A model, with its ability to effectively align entities across different KGs, has potential applications in the following fields:

**Intelligent search and question answering:** Search engines and question-answering systems rely on knowledge graphs to provide accurate and context-aware responses. However, inconsistencies in entity representations across different sources can lead to suboptimal results. ARNM-DAE2A improves search relevance and enhances answer accuracy by aligning equivalent entities, ensuring consistent information retrieval. In biomedical databases, for instance, the model helps unify different names for the same disease, facilitating more comprehensive access to medical knowledge.

**Enterprise knowledge management and data integration:** Organizations maintain multiple databases with overlapping but inconsistently labeled information. Effective entity

alignment enables seamless integration of disparate data sources, improving data consistency and decision-making. ARNM-DAE2A can assist in unifying customer records, product catalogs, or financial transactions, making it particularly valuable in industries such as healthcare, finance, and supply chain management.

**Recommendation systems:** Knowledge graphs enhance recommendation models by representing relationships between users, items, and contextual information. However, inconsistencies in entity representations across platforms can reduce recommendation accuracy. By aligning entities from multiple sources, ARNM-DAE2A enables more precise associations, improving personalization in e-commerce, media streaming, and online advertising.

**Multilingual knowledge graph integration:** Many large-scale KGs, such as DBpedia and Wikidata, contain equivalent entities in different languages without explicit mappings. ARNM-DAE2A facilitates automatic cross-lingual entity alignment, reducing manual efforts and improving interoperability between multilingual knowledge bases. This is particularly beneficial for cross-lingual information retrieval, machine translation, and global-scale semantic search.

## 7 Conclusion

This paper introduces an entity alignment neighborhood matching model called ARNM-DAE2A that combines attribute information and dual attention. This model incorporates attribute information, which was not considered in the relationship-aware neighborhood matching model for entity alignment, and introduces dual attention mechanisms to enhance the learning capabilities of GCN structures. Simulations on three cross-lingual datasets validate the effectiveness of dual attention and attribute information in entity alignment methods. In future work, we plan to explore more proactive fusion mechanisms, such as cross-modal attention or dynamic feature selection-based methods, to enable attribute embedding to adaptively adjust the weights between different entity pairs and improve the robustness of alignment. At the same time, we consider introducing dyadic attention to other association tasks, such as cross-language information retrieval or KG complementation, to expand its application areas.

## Author contributions

**Conceptualization:** Weiwei Liu, Xiong YANG.

**Funding acquisition:** Junlin Gu.

**Resources:** Weiwei Liu.

**Software:** Junlin Gu.

**Supervision:** Weiwei Liu.

**Writing – original draft:** Weiwei Liu, Xiong YANG.

**Writing – review & editing:** Weiwei Liu, Xiong YANG.

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
