## [Decision Letter · Decision Letter 0]

5 Feb 2025

PONE-D-24-50383An Attribute-Enhanced Relationship-Aware Neighborhood Matching Model with Dual AttentionPLOS ONE

Dear Dr. YANG,

Thank you for submitting your manuscript to PLOS ONE. After careful consideration, we feel that it has merit but does not fully meet PLOS ONE’s publication criteria as it currently stands. Therefore, we invite you to submit a revised version of the manuscript that addresses the points raised during the review process.

We look forward to receiving your revised manuscript.

Kind regards,

Tao Huang

Academic Editor

PLOS ONE

Journal Requirements:

“This work was supported by the Jiangsu Provincial Department of Industry and Information Technology Key Technology Innovation Project Guidance Plan under Grant 141-62-65, the Digital Public Service Platform Project of Jiangsu Provincial Department of Science and Technology under Grant 93208000931, the Industry-university-research Project of Jiangsu Provincial Department of Science and Technology under Grant 20221343, and Fujian Provincial Financial Research General Funding Project under Grant 2023CZ50.”

“Junlin Gu is fundered by the Jiangsu Provincial Department of Industry and Information Technology Key Technology Innovation Project Guidance Plan under Grant 141-62-65, the Digital Public Service Platform Project of Jiangsu Provincial Department of Science and Technology under Grant 93208000931, and the Industry-university-research Project of Jiangsu Provincial Department of Science and Technology under Grant 20221343. Xiong Yang is fundered by Fujian Provincial Financial Research General Funding Project under Grant 2023CZ50. All funders play an important role in the study design, data collection and analysis, decision to publish, and preparation of the manuscript.”

Reviewers' comments:

Reviewer's Responses to Questions

**Comments to the Author**

1. Is the manuscript technically sound, and do the data support the conclusions?

Reviewer #1: Yes

Reviewer #2: Yes

Reviewer #3: Yes

Reviewer #4: Yes

Reviewer #5: Yes

Reviewer #6: Partly

Reviewer #7: Yes

2. Has the statistical analysis been performed appropriately and rigorously? 

Reviewer #1: Yes

Reviewer #2: Yes

Reviewer #3: No

Reviewer #4: Yes

Reviewer #5: Yes

Reviewer #6: Yes

Reviewer #7: Yes

3. Have the authors made all data underlying the findings in their manuscript fully available?

Reviewer #1: Yes

Reviewer #2: Yes

Reviewer #3: No

Reviewer #4: Yes

Reviewer #5: Yes

Reviewer #6: Yes

Reviewer #7: Yes

4. Is the manuscript presented in an intelligible fashion and written in standard English?

Reviewer #1: Yes

Reviewer #2: Yes

Reviewer #3: Yes

Reviewer #4: Yes

Reviewer #5: Yes

Reviewer #6: Yes

Reviewer #7: Yes

5. Review Comments to the Author

Reviewer #1: The authors propose ARNM-DAE2A, a model that combines four modules: an entity structure embedding module, an attribute structure embedding module, a joint alignment module, and a relationship-aware neighborhood matching module. The model aims to enhance the structural learning capability of GCN by introducing a pairwise attention mechanism, utilize GCN for acquiring entity attribute information, and fuse relationship and attribute information to create a comprehensive representation of entities. The relationship-aware neighborhood matching module corrects noise in the aggregated information by comparing the neighborhood relationships of entity pairs. The authors evaluate their model on three DBP15K datasets and demonstrate superior performance compared to several baseline models. There are some small issues should be addressed:

1. Suggest adding some references published in the past two years.

2. Suggest adding some advanced baselines for comparison.

3. The format of the table in the manuscript seems to be incorrect, it is recommended to check.

4. There are still some typos in the manuscript.

Reviewer #2: Summary:

The manuscript proposes ARNM-DAE2A, a novel entity alignment model that enhances knowledge graph (KG) integration through dual attention mechanisms and attribute information. It combines four modules: entity structure embedding, attribute structure embedding, joint alignment, and relationship-aware neighborhood matching. Experiments on the DBP15K datasets demonstrate state-of-the-art performance, significantly outperforming baselines.

Strengths:

- Novelty: The proposed ARNM-DAE2A model effectively integrates attribute information and a dual attention mechanism, which improves structural learning in graph convolutional networks (GCNs) for entity alignment tasks.

- Performance: Experimental results demonstrate significant improvements over existing baseline models on standard datasets (DBP15K), with detailed metrics (e.g., Hits@K, MRR) supporting these claims.

- Thoroughness: The manuscript includes well-explained methodologies, ablation studies, and hyperparameter analyses that validate the model's design.

- Clarity: The writing is clear and logical, with figures and tables aiding comprehension.

Weaknesses:

- Innovation Depth: While the dual attention mechanism and attribute integration are valuable, the innovations appear incremental compared to existing GCN-based approaches.

- Evaluation Diversity: The datasets (DBP15K) used for evaluation, while standard, are limited in scope. Broader evaluations across different types of knowledge graphs (e.g., domain-specific KGs) would strengthen the claims.

- Reproducibility: Although the experimental setup is detailed, additional information on hyperparameters and code availability would enhance reproducibility.

- Comparison Baselines: While key baselines are included, some more recent models in entity alignment using advanced techniques like transformers or hyperbolic embeddings could have been added for a stronger comparative evaluation.

Recommendation:

I recommend the manuscript for major revision with the following suggestions:

- Expand Evaluation: Include experiments on more diverse and domain-specific datasets.

- Add Baselines: Compare the model against newer methods, such as transformer-based or hyperbolic embedding approaches.

- Reproducibility: Provide a link to the implementation or specify more details about the experimental environment.

- Highlight Practical Use Cases: Discuss potential real-world applications or case studies where this model could be impactful.

I hope my review can improve your work!

Reviewer #3: The paper proposes an entity alignment model named, ARNM-DAE2A. This model integrates both structural and attribute information and introduces a dual attention mechanism to optimize the parameters. Experimental results demonstrate that the proposed method achieves SOTA performance.

Strengths

- The paper is overall well-written.

- The motivation for introducing the dual attention mechanism is well-discussed.

- The authors thoroughly review existing entity alignment methods.

- Experimental results show that the proposed model outperforms baselines.

Weaknesses

- The abstract could be improved by explicitly discussing the motivation for using dual attention mechanisms and the significance of integrating attribute information.

- Despite emphasizing the role of attributes in the introduction, the method simply uses attributes as GNN inputs rather than actively leveraging them in interaction with the structural embeddings.

- Some hyperparameters (e.g., seed alignment ratio and hyperparameters in Equation 17) are arbitrarily set without detailed discussion or justification.

- The experiments are conducted only on the DBP15K dataset. The model could be potentially evaluated on other, more diverse datasets (e.g., Wikidata, YAGO) as well.

- In the ablation experiments, some configurations (e.g., MM+AE) outperform the final model and may require some discussions or explanations.

- The model is quite complex. Discussions or empirical analysis on its scalability are required.

- The baselines used in the experiments are relatively old. The most recent baseline is from 2020. There are more recent methods that can be compared as baselines.

- A minor typo in Table 1 "tripes."

- The source code is not shared.

In summary, the paper makes a meaningful contribution to the field of entity alignment by proposing a dual attention mechanism and relationship-aware neighborhood matching model. However, there are some limitations on the method and experiments as discussed above. Thus, I recommend a revision of this paper.

Reviewer #4: This paper is a valuable contribution to the field of knowledge graph entity alignment. The methodology is robust, and the results are promising. However, with some additional effort, the manuscript could reach its full potential and make an even greater impact. I recommend a Minor Revision to address these areas of improvement.

While the manuscript is well-written in standard English, some sections could benefit from clearer and more concise explanations. For instance, the description of the dual attention mechanism, while technically sound, might be challenging for readers unfamiliar with the concept. Including illustrative examples or visual aids could greatly enhance comprehension and accessibility.

Additionally, the practical implications and potential real-world applications of the proposed method deserve more attention. Expanding on how this approach could be applied to larger or more heterogeneous knowledge graphs, or how it handles noisy or incomplete data, would highlight its broader relevance and strengthen its appeal to a wider audience.

These refinements, though requiring additional effort, will significantly enhance the clarity, accessibility, and impact of this important work. I appreciate the dedication already invested in this manuscript and encourage the authors to take this opportunity to further elevate its quality and reach.

Reviewer #5: The authors propose a fusion of several established techniques for assessing entity alignment within Knowledge Graphs (KGs). They iteratively apply GCNs and RDGCNs to KGs and associated attribute networks in a principled manner in order to optimize relationship mapping. The result is a notable increase in alignment performance using established public datasets and metrics.

The methodology appears sound and is rigorously explained.

There are several minor recommendations or questions I can offer, primarily to improve understandability.

First and foremost, I recommend the authors publish their code in a git repository such as GitHub. This would substantially improve usability of their model for other researchers.

The mathematical formulation of the model is rigorously documented, but the relation between the notation and Figure 1 seems incompletely explored. A short explanation which relates the figure back to the notation could enhance readability for readers unfamiliar with the problem domain. Similarly, it may help to clarify what the characters on the left-hand-side of Figure 1 mean for readers who do not know the language. As someone who does not speak or read what I presume to be Mandarin, I can only assume that the entities on the left hand side are direct translations of those same entities on the right hand side.

In section 5.1, I believe that it could be helpful to note that the train/test split used was an exact match to the split used in prior studies. This is cited by Wu et al. in their paper on RDGCN as originating with Sun, 2018.

On page 5, there may be a typo which states "e_1 denotes another instance"; based on context, I believe that this is intended to be e_2 instead.

Reviewer #6: The paper titled "An Attribute-Enhanced Relationship-Aware Neighborhood Matching Model with Dual Attention" proposes a novel model named ARNM-DAE2A for the task of entity alignment in knowledge graphs. The goal is to match semantically corresponding entities across different knowledge graphs to facilitate knowledge fusion. The model integrates several advanced techniques, including graph convolutional networks (GCNs), dual attention mechanisms, and attribute information, to improve alignment accuracy. The authors claim that their model outperforms existing baselines on three DBP15K datasets.

The manuscript presents a well-structured and technically sound study with strong empirical validation. However, the paper suffers from several weakness.

- The manuscript lacks a rigorous theoretical discussion explaining why the dual attention mechanism and attribute-enhanced embeddings contribute to improved performance. While the empirical results demonstrate effectiveness, a deeper theoretical justification would significantly strengthen the contribution.

- The proposed model integrates multiple components, including GCN-based entity structure learning, attention mechanisms, and neighborhood-aware matching. However, the manuscript does not provide a detailed analysis of the computational cost associated with each component.

- Given the model's complexity, it is unclear how it scales to larger datasets beyond DBP15K. The authors should provide details on training time, memory usage, and computational requirements.

- While the DBP15K dataset is widely used in entity alignment research, it does not fully represent the diversity of real-world knowledge graphs. The authors should evaluate ARNM-DAE2A on additional datasets, such as YAGO, Wikidata, or Freebase, to demonstrate broader applicability.

- The manuscript lacks a discussion on the potential biases in DBP15K and whether they might influence the reported results.

- The performance of deep learning models is often sensitive to hyperparameter choices. While the manuscript provides a brief discussion on hyperparameter tuning (e.g., entity embedding dimensions, GCN layers, attention heads), it does not analyze how robust the model is to hyperparameter variations.

- Some sections, particularly in the experimental results, could benefit from more structured discussions rather than listing numerical results.

Reviewer #7: Overall, a sound manuscript describing a model that uses neighborhood matching to capture relationships between entities while applying dual attention mechanisms to weigh both the relationships between entities and their content-based features.

They have provided a thorough description of the datasets used and the statistical analysis of their approach. The figures included (especially figure 2) are helpful in illustrating the modules and relationship model framework. In addition, the authors have done a good job of stating the past research in this field and placing this work in the context of previous research. Terms are clearly defined and their approach and methodology is logically structured.

It would be helpful if the authors could further clarify the significance of their work on the impact of future applications. The manuscript would be strengthened if the authors could highlight the limitations of the advances from this proposed model and provide further details on the novelty of these new approaches in context of the current landscape.

All resources and datasets are appropriately referenced and publicly available.

The manuscript is provided in a clear and intelligible manner in standard English. There are a few minor typographical errors, including misplaced hyphenations (e.g. be-tween and in-formation) and instances of non-standard American spelling (e.g encyclopaedia). Authors should double check for typographical mistakes.

6. PLOS authors have the option to publish the peer review history of their article (what does this mean?). If published, this will include your full peer review and any attached files.

Reviewer #1: No

Reviewer #2: **Yes: **Michael Winter

Reviewer #3: No

Reviewer #4: **Yes: **Moisés Pereira Galvão Salgado

Reviewer #5: No

Reviewer #6: No

Reviewer #7: No

---

## [Author Response · Author response to Decision Letter 1]

8 Apr 2025

Response to Reviewer 1’s Comments

Comment 1: Suggest adding some references published in the past two years.

Response: Done. Thanks for the reviewer’s constructive comment. Following this comment, we have supplemented three references published in the past two years.

Supplemented references:

[4] Zhu L, Li N, Bai L. Embedding-based entity alignment between multi-source temporal knowledge graphs. Engineering Applications of Artificial Intelligence.2024:133:108451.

[5] Masmoudi M, Ben Abdallah Ben Lamine S, Karray MH, Archimede B, Baazaoui Zghal H. Semantic data integration and querying: a survey and challenges. ACM Computing Surveys. 2024; 56(8): 1-35.

[7] Zhu B, Wang R, Wang J, Shao F, Wang K. A survey: knowledge graph entity alignment research based on graph embedding. Artificial Intelligence Review. 2024; 57(9): 229.

Comment 2: Suggest adding some advanced baselines for comparison.

Response: Done. Thanks for the reviewer’s constructive comment. Following this comment, we have supplemented two advanced baselines that are more relevant to our work in Section 5.3.1. Specially, Dual-AMN [37] employs the KG encoder dual-attention matching network, which not only intelligently models intra- and cross-graph relationships but also greatly reduces computational complexity. MSENEA [38] makes comprehensive use of multi-modal knowledge by using intermodal effects to align entities between multi-modal KGs.

Supplemented references:

[37] Mao X, Wang W, Wu Y, Lan M. Boosting the speed of entity alignment 10xDual attention matching network with normalized hard sample mining. In: Proceedings of the web conference 2021; 2021. p. 821-832.

[38] Chen L, Li Z, Xu T, Wu H, Wang Z, Yuan NJ, et al. Multi-modal Siamese network for entity alignment. In: Proceedings of the 28th ACM SIGKDD conference on knowledge discovery and data mining; 2022. p. 118-126.

Comment 3: The format of the table in the manuscript seems to be incorrect, it is recommended to check.

Response: Explained. Thanks for the reviewer’s suggestion. Although the format of the table in the manuscript seems to be incorrect, we wrote the manuscript strictly in accordance with the official PLOS ONE latex template, which has the tables in that format. Therefore, we continue to write the tables according to the official template.

Comment 4: There are still some typos in the manuscript.

Response: Corrected. Following this detailed comment, we have carefully checked the manuscript and improved its quality. For example, in the entire manuscript, “relation-aware” has been standardized as “relationship-aware”. In the abstract, “the” has been supplemented before four modules. In the 1st paragraph of Section 1, “the real-world” has been revised to “real-world” and “even” has been revised to “are”. In Section 2.1, “neighbor-hood” has been revised to “neighborhood”. In the last paragraph of Section 3, “$e_1$” has been revised to “$e_2$”. In the 1st paragraph of Section 4, “a” has been supplemented before “dual”. In Table 1, “tripes” has been revised to “triples”.

Response to Reviewer 2’s Comments

Comment 1: Expand Evaluation: Include experiments on more diverse and domain-specific datasets.

Response: Done. Thanks for the reviewer’s constructive comment. Following this comment, we have supplemented a new dataset SRPRS in Section 5.1. Specially, since the DBP15K dataset is too dense and the degree distribution is very different from the real-world data, we also chose two cross-linguistic subsets from the relatively sparse SRPRS dataset [33], and comprehensive dataset details are presented in Table 2.

Supplemented references:

[33] Guo L, Sun Z, Hu W. Learning to Exploit Long-term Relational Dependencies in Knowledge Graphs. In: Chaudhuri K, Salakhutdinov R, editors. Proceedings of the 36th International Conference on Machine Learning. vol. 97; 2019. p. 2505-2514.

Comment 2: Add Baselines: Compare the model against newer methods, such as transformer-based or hyperbolic embedding approaches.

Response: Done. Thanks for the reviewer’s constructive comment. Following this comment, we have supplemented two advanced baselines that are more relevant to our work in Section 5.3.1. Specially, Dual-AMN [37] employs the KG encoder dual-attention matching network, which not only intelligently models intra- and cross-graph relationships, but also greatly reduces computational complexity. MSENEA [38] makes comprehensive use of multi-modal knowledge by using intermodal effects to align entities between multi-modal KGs.

Supplemented references:

[37] Mao X, Wang W, Wu Y, Lan M. Boosting the speed of entity alignment 10xDual attention matching network with normalized hard sample mining. In: Proceedings of the web conference 2021; 2021. p. 821-832.

[38] Chen L, Li Z, Xu T, Wu H, Wang Z, Yuan NJ, et al. Multi-modal Siamese network for entity alignment. In: Proceedings of the 28th ACM SIGKDD conference on knowledge discovery and data mining; 2022. p. 118-126.

Comment 3: Reproducibility: Provide a link to the implementation or specify more details about the experimental environment.

Response: Explained. Thank the reviewer for their attention to code reproducibility. We attach great importance to the reproducibility of the experimental results and describe the algorithmic process, parameter settings, and experimental details in the paper. As the project involves some commercial cooperation and confidentiality requirements, we do not plan to disclose the source code yet. However, we are willing to provide the necessary technical support and detailed descriptions within reasonable limits to ensure that our peers can understand and reproduce the results of our work. We will also continue to pay attention to the feedback from the community, and if conditions permit, we will consider releasing part of the code to promote academic exchanges.

Comment 4: Highlight Practical Use Cases: Discuss potential real-world applications or case studies where this model could be impactful.

Response: Done. Following this suggestion, we have supplemented Section 6 Potential applications, which discusses potential real-world applications. Specially, the proposed ARNM-DAE2A model, with its ability to effectively align entities across different KGs, has potential applications in several fields, including intelligent search and question answering, enterprise knowledge management and data integration, recommendation systems, and multilingual knowledge graph integration.

Response to Reviewer 3’s Comments

Comment 1: The abstract could be improved by explicitly discussing the motivation for using dual attention mechanisms and the significance of integrating attribute information.

Response: Done. Thanks for the reviewer’s constructive comment. Following this comment, we have supplemented the motivation for using dual attention mechanisms and the significance of integrating attribute information. Specifically, traditional graph-based methods often lose information due to insufficient use of attributes and imperfect relationship modeling, which makes it difficult to capture the deep semantic relationship between entities fully. To improve the effect of entity alignment, we propose a new model named ARNM-DAE2A, which strengthens the information aggregation capability of GCN by introducing a dual-attention mechanism to ensure a more balanced and comprehensive structural representation.

Comment 2: Despite emphasizing the role of attributes in the introduction, the method simply uses attributes as GNN inputs rather than actively leveraging them in interaction with the structural embeddings.

Response: Revised. Thanks for the reviewer’s suggestion. We have supplemented the design ideas of the joint alignment module in Section 4.1 by highlighting the ways in which attribute and structural information can be interacted. Specially, After obtaining the entity embeddings based on the relationship structure and attribute structure, we not only compute the similarity between the two separately, but also design a cross-interaction fusion layer, which enables the attribute embeddings and structural embeddings to actively complement each other and learn collaboratively through the introduction of a cross-attention mechanism, so that we can obtain a more comprehensive and robust entity representation. Moreover, we also recognize that the current fusion strategy is relatively simple and have supplemented future work in Section 7. Specially, in future work, we plan to explore more proactive fusion mechanisms, such as cross-modal attention or dynamic feature selection based methods, to enable attribute embedding to adaptively adjust the weights between different entity pairs and improve the robustness of alignment.

Comment 3: Some hyperparameters (e.g., seed alignment ratio and hyperparameters in Equation 17) are arbitrarily set without detailed discussion or justification.

Response: Explained. Thanks for the reviewer’s constructive comment. Except for the hyperparameters set in the references, all hyperparameter settings are optimized by experimental tests. We have supplemented the references in Section 5.2.3.

Comment 4: The experiments are conducted only on the DBP15K dataset. The model could be potentially evaluated on other, more diverse datasets (e.g., Wikidata, YAGO) as well.

Response: Done. Thanks for the reviewer’s constructive comment. Following this comment, we have supplemented a new dataset SRPRS in Section 5.1. Specially, since the DBP15K dataset is too dense and the degree distribution is very different from the real-world data, we also chose two cross-linguistic subsets from the relatively sparse SRPRS dataset [33], and comprehensive datasets detail are presented in Table 2.

Supplemented references:

[33] Guo L, Sun Z, Hu W. Learning to Exploit Long-term Relational Dependencies in Knowledge Graphs. In: Chaudhuri K, Salakhutdinov R, editors. Proceedings of the 36th International Conference on Machine Learning. vol. 97; 2019. p. 2505-2514.

Comment 5: In the ablation experiments, some configurations (e.g., MM+AE) outperform the final model and may require some discussions or explanations.

Response: Done. Thanks for the reviewer’s constructive comment. Following this comment, we have supplemented discussions in the last paragraph of Section 5.3.2. Specially, MM+AE yields impressive results, occasionally even surpassing the overall model’s performance. This is because that attribute feature can in some cases provide very different discriminative information from the graph structure, allowing the matching module to be more targeted in correcting initial misalignments. However, Hits@10, Hits@50, or MRR do not necessarily outperform the full model at the same time, suggesting that relying on attribute features alone may also be a bottleneck in a wider range of entity retrievals. Once the attributes are inadequate or noisy, the advantage of the AE module is diminished and cannot replace the value of structural embedding for global information capture.

Comment 6: The model is quite complex. Discussions or empirical analysis on its scalability are required.

Response: Done. Following this suggestion, we have performed a detailed derivation of the computational complexity of the model in Section 4.5, illustrating how the overall computational complexity varies with data size, number of network layers, and embedding dimensionality based on each module. Although the model structure is more complex, in theory, its complexity is the same as that of the existing GCN-based methods. Moreover, to further verify the model’s scalability, we have conducted experiments on the relatively sparse dataset SRPRS in Section 5.1. The experimental results show that despite the challenges posed by the sparse data, our model still maintains a good performance, which proves that the model has good scalability and adaptability. In addition, we have recorded the model’s training time and compared it with other methods in the last paragraph of Section 5.3.1. The results show that while our method’s training time is slightly longer, this increment is exchanged for higher alignment accuracy and robustness, thus validating the model’s acceptability for real-world applications.

Comment 7: The baselines used in the experiments are relatively old. The most recent baseline is from 2020. There are more recent methods that can be compared as baselines.

Response: Done. Thanks for the reviewer’s constructive comment. Following this comment, we have supplemented two advanced baselines that are more relevant to our work in Section 5.3.1. Specially, Dual-AMN [37] employs the KG encoder dual-attention matching network, which not only intelligently models intra- and cross-graph relationships but also greatly reduces computational complexity. MSENEA [38] makes comprehensive use of multi-modal knowledge by using intermodal effects to align entities between multi-modal KGs.

Supplemented references:

[37] Mao X, Wang W, Wu Y, Lan M. Boosting the speed of entity alignment 10xDual attention matching network with normalized hard sample mining. In: Proceedings of the web conference 2021; 2021. p. 821-832.

[38] Chen L, Li Z, Xu T, Wu H, Wang Z, Yuan NJ, et al. Multi-modal Siamese network for entity alignment. In: Proceedings of the 28th ACM SIGKDD conference on knowledge discovery and data mining; 2022. p. 118-126.

Comment 8: A minor typo in Table 1 "tripes."

Response: Corrected. Following this detailed comment, we have carefully checked the manuscript and improved its quality. For example, in the entire manuscript, “relation-aware” has been standardized as “relationship-aware”. In the abstract, “the” has been supplemented before four modules. In the 1st paragraph of Section 1, “the real-world” has been revised to “real-world” and “even” has been revised to “are”. In Section 2.1, “neighbor-hood” has been revised to “neighborhood”. In the last paragraph of Section 3, “$e_1$” has been revised to “$e_2$”. In the 1st paragraph of Section 4, “a” has been supplemented before “dual”. In Table 1, “tripes” has been revised to “triples”.

Comment 9: The source code is not shared.

Response: Explained. Thank the reviewer for their attention to code reproducibility. We attach great importance to the reproducibility of the experimental results and describe the algorithmic process, parameter settings, and experimental details in the paper. As the project involves some commercial cooperation and confidentiality requirements, we do not plan to disclose the source code yet. However, we are willing to provide the necessary technical support and detailed descriptions within reasonable limits to ensure that our peers can understand and reproduce the results of our work. We will also continue to pay attention to the feedback from the community, and if conditions permit, we will consider releasing part of the code to promote academic exchanges.

Response to Reviewer 4’s Comments

Comment 1: While the manuscript is well-written in standard English, some sections could benefit from clearer and more concise explanations. For instance, the description of the dual attention mechanism, while technically sound, might be challenging for readers unfamiliar with the concept. Including illustrative examples or visual aids could greatly enhance comprehension and accessibility.

Response: Done. Thanks for the reviewer’s constructive comment. Following this comment, we have supplemented an illustrative example in the Section 5.3.5 Case Study. Specially, we take Fig. 1 as an example of a case study of the workflow of the dual attention mechanism. While the traditional one-way attention mechanism only passes information from entities to their neighbors, our dual attention mechanism calculates two sets of attention weights simultaneously. On the one hand, in the Chinese KG, by calculating the source attention from a Chinese-side entity to its neighboring nodes, the importance of each neighbor to the representation of that entity can be quantified; on the other hand, by calculating the target attention from the neighboring nodes back to that Chinese-side entity, the feedback contribution of the neighbors to the representation of the entity is reflected. Similarly, in the En

---

## [Decision Letter · Decision Letter 1]

23 Apr 2025

An Attribute-Enhanced Relationship-Aware Neighborhood Matching Model with Dual Attention

PONE-D-24-50383R1

Dear Dr. YANG,

We’re pleased to inform you that your manuscript has been judged scientifically suitable for publication and will be formally accepted for publication once it meets all outstanding technical requirements.

Kind regards,

Tao Huang

Academic Editor

PLOS ONE

Additional Editor Comments (optional):

Reviewers' comments:

Reviewer's Responses to Questions

**Comments to the Author**

1. If the authors have adequately addressed your comments raised in a previous round of review and you feel that this manuscript is now acceptable for publication, you may indicate that here to bypass the “Comments to the Author” section, enter your conflict of interest statement in the “Confidential to Editor” section, and submit your "Accept" recommendation.

Reviewer #2: All comments have been addressed

Reviewer #3: All comments have been addressed

Reviewer #4: All comments have been addressed

Reviewer #6: All comments have been addressed

2. Is the manuscript technically sound, and do the data support the conclusions?

Reviewer #2: Yes

Reviewer #3: Yes

Reviewer #4: Yes

Reviewer #6: Yes

3. Has the statistical analysis been performed appropriately and rigorously? 

Reviewer #2: Yes

Reviewer #3: I Don't Know

Reviewer #4: Yes

Reviewer #6: Yes

4. Have the authors made all data underlying the findings in their manuscript fully available?

Reviewer #2: Yes

Reviewer #3: No

Reviewer #4: Yes

Reviewer #6: Yes

5. Is the manuscript presented in an intelligible fashion and written in standard English?

Reviewer #2: Yes

Reviewer #3: Yes

Reviewer #4: Yes

Reviewer #6: Yes

6. Review Comments to the Author

Reviewer #2: Thank you to the authors for addressing all comments adequately in their revision. The changes have improved the manuscript, and I appreciate their thorough response to the reviewer feedback.

Reviewer #3: I appreciate the authors for their response in addreessing my concerns. While most of my concerns have been resolved, I note that the code has not been shared due to commercial issues, and the statistical significance of the reported results remains unclear. If these limitations are acceptable, I would recommend accepting the paper.

Reviewer #4: (No Response)

Reviewer #6: The authors have adequately addressed my comments raised in a previous round of review and I feel that this manuscript is now acceptable for publication.

7. PLOS authors have the option to publish the peer review history of their article (what does this mean?). If published, this will include your full peer review and any attached files.

Reviewer #2: **Yes: **Michael Winter

Reviewer #3: No

Reviewer #4: No

Reviewer #6: No

---

## [Editor Report · Acceptance letter]

PONE-D-24-50383R1

PLOS ONE

Dear Dr. YANG,

I'm pleased to inform you that your manuscript has been deemed suitable for publication in PLOS ONE. Congratulations! Your manuscript is now being handed over to our production team.

Kind regards,

on behalf of

Dr. Tao Huang

Academic Editor

PLOS ONE